# Self-powered ultraflexible photonic skin for continuous bio-signal detection via air-operation-stable polymer light-emitting diodes

Hiroaki Jinno [1,2,5], Tomoyuki Yokota [1], Mari Koizumi[1], Wakako Yukita[1], Masahiko Saito[3], Itaru Osaka[3], Kenjiro Fukuda [2,4] & Takao Someya [1,2,4✉]

Ultraflexible optical devices have been used extensively in next-generation wearable electronics owing to their excellent conformability to human skins. Long-term health monitoring also requires the integration of ultraflexible optical devices with an energy-harvesting power source; to make devices self-powered. However, system-level integration of ultraflexible optical sensors with power sources is challenging because of insufficient air operational stability of ultraflexible polymer light-emitting diodes. Here we develop an ultraflexible self-powered organic optical system for photoplethysmogram monitoring by combining air-operation-stable polymer light-emitting diodes, organic solar cells, and organic photodetectors. Adopting an inverted structure and a doped polyethylenimine ethoxylated layer, ultraflexible polymer light-emitting diodes retain 70% of the initial luminance even after 11.3 h of operation under air. Also, integrated optical sensors exhibit a high linearity with the light intensity exponent of 0.98 by polymer light-emitting diode. Such self-powered, ultraflexible photoplethysmogram sensors perform monitoring of blood pulse signals as 77 beats per minute.

[1] Electrical and Electronic Engineering and Information Systems, The University of Tokyo, Tokyo, Japan. [2] Center for Emergent Matter Science, RIKEN, Saitama, Japan. [3] Department of Applied Chemistry, Graduate School of Engineering, Hiroshima University, Hiroshima, Japan. [4] Thin-film Device Laboratory, RIKEN, Saitama, Japan. [5] Present address: Nanomaterials Engineering Research Group, ETH zürich, Zürich, Switzerland. ✉email: someya@ee.t.u-tokyo.ac.jp

Organic semiconducting devices have been used extensively in next-generation wearable electronics because of their light-weight, thinness, and flexibility[1–3]. Flexible displays consisting of organic light-emitting diodes (LEDs) have been installed in smartwatches and wristband-type applications, contributing to a reduction in power consumption[4]. In addition, an all-organic optoelectronic sensor has been developed for pulse oximetry, by integrating organic LEDs and organic photodetectors[5]. Imparting flexibility to such organic optical sensors enables long-term monitoring of health conditions with reduced discomfort when these sensors are attached directly on human skin.

Recent advances in the field of flexible optical integrated devices have made these devices favourable for optical sensor applications, such as photoplethysmogram (PPG)[6–10], pulse oximetry[10–12]. High-resolution imaging devices have been used in advanced applications, such as vein authentication[13,14], finger-print imaging[13,15], and X-ray imaging[16,17]. Self-powered systems such as thermoelectric conversion devices[18] and organic solar cells[19] have been developed by integrating organic optical sensors and flexible energy-harvesting devices. Another approach involves combining organic LEDs with battery modules to develop a patch system for photobiomodulation that can be used for wound healing[20].

In addition to integrated devices, devices with improved conformability to biological tissues are necessary for stable long-term monitoring of biological signals. Conformability can be achieved by combining two approaches: using soft materials having Young's moduli similar to biological tissues, and reducing total device thickness[21]. Thanks to smaller Young's moduli of organic materials compared to inorganic materials, and the compatibility of organic devices to low-temperature and solution processes, the total thickness can be reduced to 1–2 μm without sacrificing the device performance leading to the reduction of discomfort on the skin[22–25]. An ultrathin organic optical sensor has been developed with several microns thick[26–29].

However, system-level integration of such thin organic optical sensors with power sources remains challenging. A significant obstacle to this type of integration is the insufficient operational stability of ultrathin organic LEDs under ambient air conditions[26–28], which hinders the long-term monitoring of biological signals.

Here we propose an ultrathin self-powered organic optical system for PPG monitoring. This system consists of three ultra-thin electronic devices: polymer light-emitting diodes (PLEDs), organic solar cells, and organic photodetectors. By adopting an inverted structure and a polyethylenimine ethoxylated (PEIE) layer doped with 8-quinolinolato lithium (Liq)[30] as the electron-transport layer (ETL), organic LEDs exhibit improved operation stability under ambient air conditions. Because of the intrinsic air operational stability of inverted PLED with PEIE:Liq electron injection layers, the fabricated ultraflexible PLED with no passivation retains 70% of its initial luminance lifetime of 11.3 h under ambient air, which is more than three times the luminance lifetime of conventional ultraflexible PLEDs. Integrated optical sensor exhibits a high linearity with the light intensity exponent of 0.98 by PLED light source. Such self-powered, ultraflexible PPG sensor shows a blood pulse monitoring during 7 s and can detect blood pulse rate of 77 beats per minute (bpm) on human hands.

## Results

### Air-operation-stable PLEDs
Figure 1a and Supplementary Fig. 1 show a schematic and photographs of ultraflexible, self-powered, PPG sensor operated on a human hand. The PPG sensors consisted of two types of ultraflexible devices: an ultraflexible PLED

and ultraflexible organic photodiodes (OPDs) (Supplementary Fig. 2). The ultraflexible PPG sensors and organic photovoltaic (OPV) modules were connected with flexible gold wiring, which was a 100-nm-thick gold electrode fabricated on a 12.5-μm-thick polyimide film (Fig. 1b). A top-view image of an OPV module and a PPG sensor are shown in Fig. 1c, d, respectively. Figure 1e shows a schematic of the electrical circuit of a self-powered PPG device. The PLEDs were powered by ultraflexible OPV modules, and PPG data obtained from the OPD was transferred to the oscilloscope. The PLEDs were fabricated on a 1.5-μm-thick flexible substrate made with a 1-μm-thick parylene substrate and a 500-nm-thick planarisation layer of SU-8. The PLED comprised stacked layers of Indium tin oxide (ITO)/Zinc oxide (ZnO)/PEIE:Liq/Superyellow (SY)/MoO$_X$/Al (Fig. 2a).

A polymer of SY was used as the emission layer, providing highly efficient light emission[31]. To achieve air-stable operation of the PLED, an inverted structure was introduced with an electron injection layer of Liq-doped PEIE. While the inverted structure allows the use of an air-stable cathode in PLEDs and improves the air stability of PLEDs[32], doping PEIE with Liq can significantly improve the operational stability of PLEDs. To optimise the doping concentration of PEIE with Liq, the effect of the Liq wt% on the inverted PLED was examined (Supplementary Fig. 3). The J–L–V characteristics of the PLED show comparable electrical and optical outputs (>10$^4$ cd m$^{-2}$ at 10 V of applied voltage) for each doping concentration (Supplementary Fig. 3a, b). As shown in Supplementary Fig. 3c, 30 wt% of Liq was observed to be an optimal ratio in terms of current efficiency. For the air-operation stability, 10 and 30 wt% of Liq show longer lifetime to be half of initial luminance (Supplementary Fig. 3d). Supplementary Fig. 3e summarises the variations in the efficiency and lifetime of the PLED with varying doping concentrations of Liq in the PEIE layer. From these results, we concluded that 30 wt% is the optimal Liq doping concentration to achieve an air-operation-stable inverted PLED. Therefore, the PEIE layer doped with 30 wt% Liq was used as the ETL for successive experiments. While the WF of PEIE:Liq shifts to the shallower level until the concentration of Liq up to 50 wt%, WF shifts back to the deeper level after the concentration of Liq over 50%. This WF shifts correspond to the electron injection of the PLED, which is improved by WF shifts to the shallower level. As a result of WF shift, 30 wt% showed the best result of efficiency[30]. The device stabilities under constant 8-V operation in ambient air were compared between various reported structures and ETLs (Supplementary Fig. 4). As a conventional structure, ITO/Poly(3,4-ethylenedioxythiophene)-poly(styrenesulfonate) (PEDOT:PSS)/SY/NaF/Al was used. As inverted structures with different ETLs, ITO/ZnO/SY/MoO$_X$/Al, ITO/ZnO/Poly-ethylenimine (PEI)/SY/MoO$_X$/Al, and ITO/ZnO/PEIE:Liq/SY/MoO$_X$/Al were compared. While the inverted structure with PEI layers initially showed an efficient luminance of 5 cd A$^{-1}$, its air stability became lower than that of the structure without PEI layers. The conventional structure showed a relatively stable luminance for up to 200 min, after which its luminance decreased abruptly. The PEIE:Liq layer showed the best air-stability among all the examined PLED structures. A comparison of the current efficiencies of various inverted-structure PLEDs is shown in Supplementary Fig. 5. We chose F8BT as a conventional emission layer polymer and PEI as a conventional ETL polymer. When the current efficiency of the inverted-structure PLED with F8BT and PEI reached 3.4 cd A$^{-1}$, the current efficiencies of the inverted-structure PLED with SY and PEI and that with SY and PEIE reached 6.9 and 11.7 cd A$^{-1}$, respectively. The operational lifetime and turn-on voltage of the inverted PLED with PEIE:Liq and SY were compared to those in previous studies (Fig. 2b)[28,33–45]. The inverted PLED has an excellent half luminance lifetime of 41.5 h with 8-V operation in

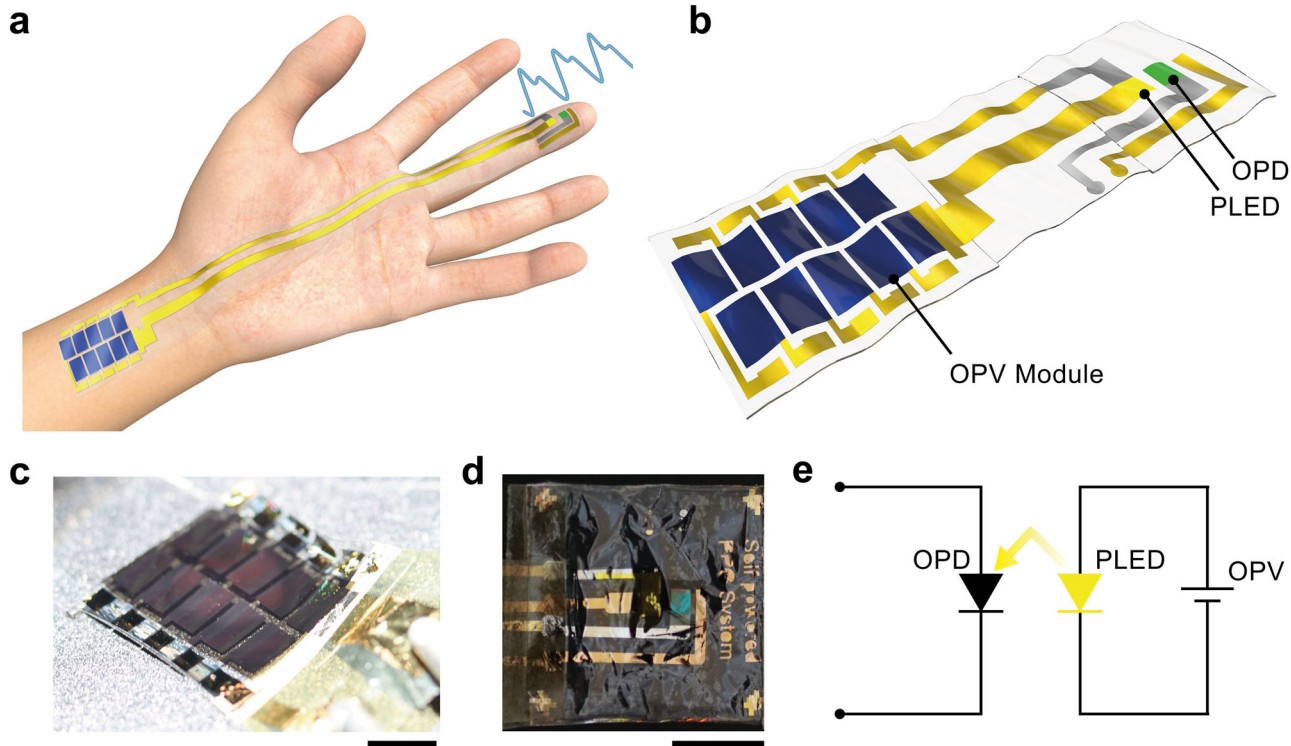

**Fig. 1 Ultraflexible, self-powered photoplethysmogram sensor. a** Schematic diagram of the ultraflexible, self-powered photoplethysmograpm (PPG) sensor on human hands. Ultraflexible organic photovoltaic (OPV) module generates electrical power from sunlight and drives polymer light-emitting diode (PLED) and organic photodiode (OPD). **b** Schematic diagram of the self-powered PPG sensor with PLED, OPD, and OPV module. **c** A photograph of the ultraflexible OPV module under one-sun illumination. Scale bar indicates 1 cm. **d** A top-view photograph of the ultraflexible, self-powered PPG sensor. Scale bar indicates 1 cm. **e** An electrical circuit of self-powered PPG sensor.

ambient air without encapsulation, while maintaining a low turn-on voltage (4.7 V). Turn-on voltage is determined as the voltage at which the luminance reaches to more than 1000 cd m$^{-2}$. Also, their angular distribution follows with Lambert's cosine law, which means our PLED has almost ideal diffusely reflecting surface (Supplementary Fig. 6).

After the characteristics of the inverted PLED on the glass substrate were examined, the inverted PLED on the ultraflexible substrate was fabricated. When the PLED was fabricated on an ultraflexible substrate, it showed a turn-on voltage of 4.7 V, which is equal to that of the PLED on the glass substrate (4.7 V) (Fig. 2c). Figure 2d shows the ultraflexible inverted PLED during 6-V operation in air. As shown in Fig. 2e, the ultraflexible PLED exhibits a current efficiency of 14 cd A$^{-1}$ at a current density of 24 mA cm$^{-2}$; this efficiency is comparable to or even higher than that of the PLED on the glass substrate. The external quantum efficiency (EQE) of PLED on ultraflexible substrate also shows comparable value to that on glass substrate (Supplementary Fig. 7). The mechanical durability of ultraflexible PLED was evaluated with the cyclic bending test. In the cyclic bending test, the ultraflexible PLED was bent cyclically to the bending radius of 5 mm. The results of the bending test is shown in Supplementary Fig. 8. Even after the device was bent cyclically with 50 times, the J–V curves did not show any degradation (Supplementary Fig. 8). Also, the luminance of PLED still remains 982 cd/m$^2$ which is comparable value as that of the initial value (918 cd/m$^2$) after the 50 cyclic bending (Supplementary Fig. 8b). Finally, to evaluate the operational stability under ambient air, the operational lifetimes of the PLEDs on both substrates under ambient air were evaluated under a constant current of 8 mA cm$^{-2}$ (Fig. 2f). The inverted PLED on the glass substrate showed lifetime for 70% of initial luminance as 14.7 h. In the inverted PLED encapsulated

with a 1-µm-thick parylene layer, the air-operation stability of the PLED improved to 33.4 h. When the PLED was fabricated on the ultraflexible substrate, the lifetime for 70% of its initial luminance decreased to 11.3 h. This is because of the low gas barrier property of ultraflexible substrates compared with that of glass[46]. Although the lifetime of the ultraflexible PLED was lower than that of the PLED fabricated on glass, it was still more than three times to the lifetime of a conventional ultraflexible PLED, which is 4 h[28]. To assess the reason of non-diode behaviour of our ultraflexible PLED (Fig. 2c), surface morphology of the PLED interface layer was studied by using atomic force microscope (AFM) (Supplementary Fig. 9). In the AFM images, the morphological difference clearly observed between bare ITO surface and ZnO interface layer (Supplementary Fig 9a, b), which shows surface roughness increase in ZnO layer. Moreover, ZnO interface layer on ultraflexible substrates shows even higher roughness compared to that on glass substrate (Supplementary Fig. 9b, c), which might be one of the reasons of non-diode behaviour of the ultraflexible PLED. However, addition of Liq into ZnO layer does not affect morphology dramatically (Supplementary Fig. 9b, d). These results proof us that the ZnO layer is a main contribution for increasing roughness, which would result in the non-diode behaviour of our ultraflexible inverted PLED.

**Ultraflexible organic photodetectors**. For an ultraflexible OPD, a blended film of 2,5-bis(3-(2-ethylhexyl-5-(trimethylstannyl)thio-phen-2-yl)thiazolo[5,4-d]thiazole-2-butyloctyl (PTzNTz-BOBO) and[6,6]-Phenyl-C71-butyric acid methyl ester (PC$_{71}$BM) is used as the active layer. A stack of ITO/ZnO/PTzNTz-BOBO:PC$_{71}$BM/ MoOx/Ag was chosen as the OPD structure in this study (Fig. 3a). To use OPD as an application of PPG sensors, linearity factors

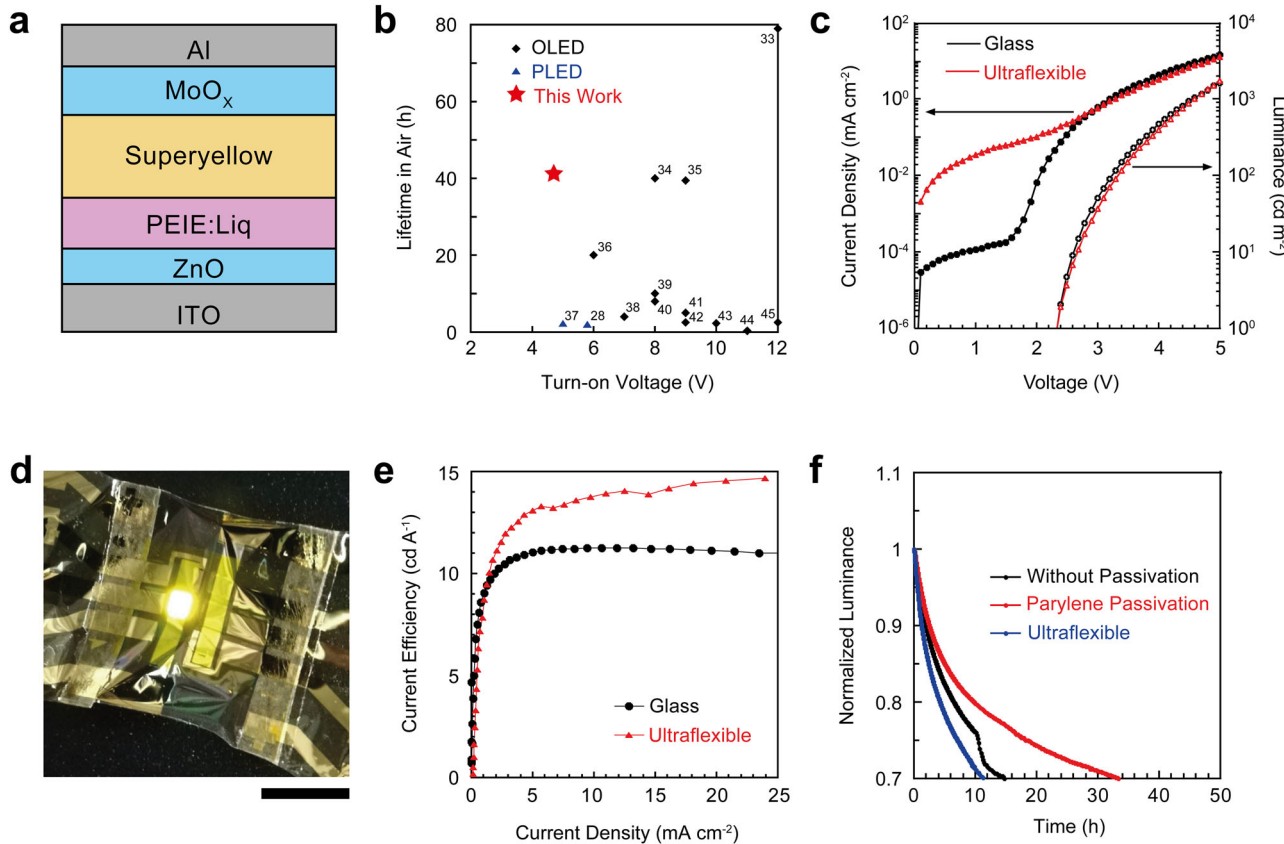

**Fig. 2 Air operational stable, ultraflexible polymer light-emitting diodes. a** Schematic diagram of cross-sectional view of the stack present in polymer light-emitting diode (PLED). The device structure is Indium tin oxide (ITO)/Zinc oxide (ZnO)/Polyethyleneimine ethoxylated (PEIE): 8-quinolinolato lithium (Liq)/Superyellow/Molybdenum oxide ($MoO_X$)/Al. **b** Comparison of the lifetime without encapsulation and turn-on voltage of previously reported organic light-emitting diode (OLED), which are represented with black diamonds, and PLED, with blue upward triangles. **c** Current density-luminance-voltage curves of PLED on glass (black curves) and ultraflexible substrate (red curves). **d** A photograph of the ultraflexible PLED during operation. Operation voltage of PLED was 6 V. Scale bar indicates 1 cm. **e** Current efficiency characteristics of the PLED on glass (black curves) and ultraflexible substrate (red curves). **f** Air operational stability of the PLED under constant-current operation in ambient air. Current density to drive PLED was set to be 8 mA/cm². The black line indicates the PLED on the glass substrate without any encapsulation; the red line indicates the PLED on the glass substrate with parylene encapsulation; and the blue line indicates the PLED on the ultraflexible substrate with parylene encapsulation. The ultraflexible PLEDs were measured in the freestanding state.

determined by using the input light intensity ($L$), short-circuit current ($J_{SC}$), logarithmic $L$, and open-circuit voltage ($V_{OC}$) are required[13]. Thus, the linearity of the OPD was examined for several light sources, such as simulated sunlight and light emitted from PLEDs. The dependence of the light intensity on $J_{SC}$ and $V_{OC}$ for the OPD, with simulated sunlight, is shown in Fig. 3b. The red dashed line and blue dashed line in Fig. 3b represent linearly fitted lines for $L$ and $J_{SC}$ and for $L$ and log ($V_{OC}$), respectively. The $J_{SC}$ of OPD and the L of simulated sunlight were fitted with the equation $J_{SC} = A \times L^{\alpha}$, in which A is a constant value determined using the power-conversion efficiency (PCE) of the OPD. α is the light intensity exponent, which indicates the linearity and accuracy of the OPD's response to the input light intensity. The α calculated from the experimental result was 0.95 for sunlight. The obtained value of α for sunlight was excellent because the value was quite close to the theoretical limit (1) of linearity of the OPD. Additionally, the EQE of the OPD at wavelengths of 300–900 nm was measured (Fig. 3c). The electroluminescence (EL) spectra of the ultraflexible inverted PLED are also shown in Fig. 3c. The EQE results represent broad light conversion of the OPD from 300 to 800 nm. The EL spectra of PLED had a peak wavelength of 590 nm and the width ranging from 500 to 800 nm. These results prove that the EQE spectra of the OPD and the EL spectra of the PLED are considerably

overlapped over a range of wavelengths; thus, the fabricated OPD is capable of efficiently absorbing the light emitted from the ultraflexible PLED. The EL wavelength of our PLED has favourable properties to PPG because of their penetration depth. Since our body tissues have a greater absorption in yellow/green wavelength[47], the penetration depth of yellow/green light is shorter compared to that of infrared. Thus, the yellow/green light will suppress the artifact of the reflection and scattering from deeper tissue and leads advantageous light properties for reflection-mode PPG[48–51]. To ensure efficient and accurate PLED light detection, the linearity of the OPD with the input PLED light was examined, as shown in Fig. 3d, e. L and $J_{SC}$ were fitted well with the same equation as used in the sunlight experiment. The α with PLED light was calculated as 0.98 which proves that our OPD can accurately measure the light intensity of the PLED. The frequency response of the OPD was examined (Supplementary Fig. 10a). The rising time and falling time were calculated as 1 ms, respectively, which is fast enough to measure blood pulse with the frequency around 1 Hz (Supplementary Fig, 10b, c). The mechanical durability of ultraflexible OPD was evaluated with the cyclic bending test. Even after the device was bent cyclically with 150 times, the $J$–$V$ curves did not show any degradation as a difference in the current density of light state and dark state (Supplementary Fig. 11a). The $J_{SC}$ difference between light state

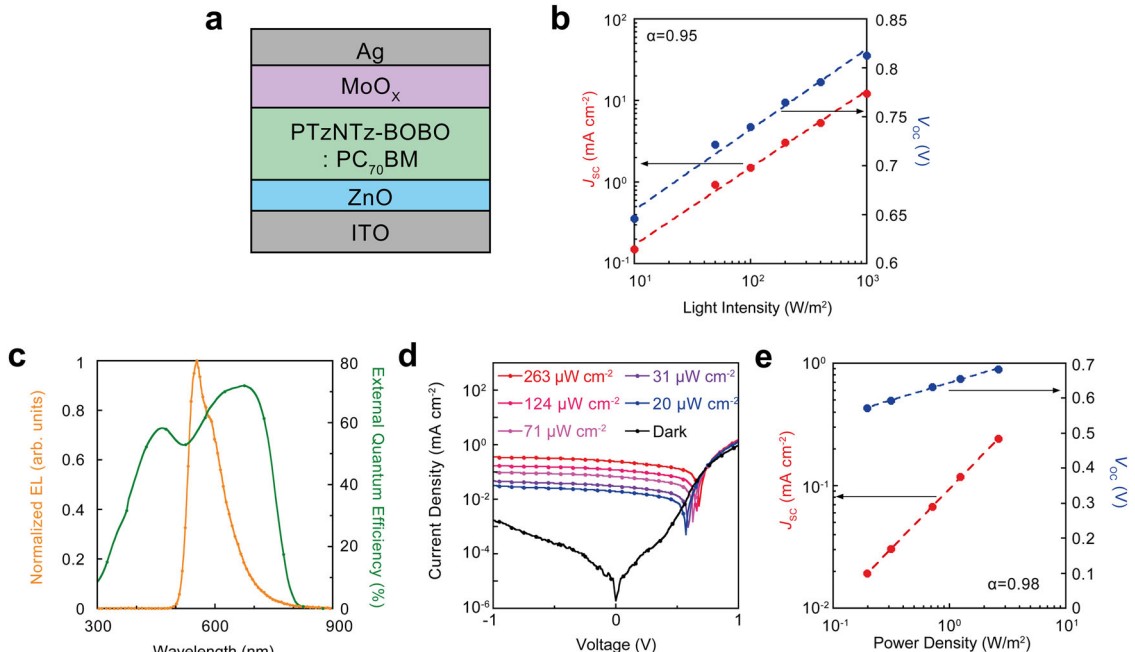

**Fig. 3 Optical responses and linearities of organic photodiode for various light sources. a** Schematic diagram of the cross-sectional view of the stack of organic photodiode (OPD). The device structure is Indium tin oxide (ITO)/Zinc oxide (ZnO)/2,5-bis(3-(2-ethylhexyl-5-(trimethylstannyl)thiophen-2-yl) thiazolo[5,4-d]thiazole-2-butyloctyl (PTzNTz-BOBO):[6,6]-Phenyl-C71-butyric acid methyl ester (PC$_{71}$BM)/Molybdenum oxide (MoO$_X$)/Ag. **b** Light intensity dependence of OPD with 1-sun illumination form solar simulator. Light intensity of solar simulator was varied from 0.01 to 1-sun with optical filters. The red dots represent short-circuit current, and the blue dots open-circuit voltage. The red dash line represents a linear fitted line of short-circuit current and light intensity. The blue dash line represents a logarithmically fitted line of open-circuit voltage and light intensity. α represents light intensity exponent of OPD with 1-sun illumination. **c** External quantum efficiency (EQE) spectra of OPD and Electroluminescence (EL) spectra of polymer light-emitting diode (PLED). The green curve represents EQE spectra of OPD, and the orange curve represents EL spectra of PLED. **d** J–V characteristics of OPD illuminated by PLED. Optical power of PLED was varied by changing driving voltage of PLED. Each of the devices are stacked together with a face-to-face configuration. The black curve represents current density-voltage dependence of OPD under dark condition, and the bule curves to red curves represent current density-voltage dependences by increasing optical power of PLED. **e** Light intensity dependence of OPD with PLED illumination. The red dots represent short-circuit current, and the blue dots open-circuit voltage. The red dash line represents a linear fitted line of short-circuit current and light intensity. The blue dash line represents a logarithmically fitted line of open-circuit voltage and light intensity. α represents light intensity exponent of OPD with PLED illumination.

and dark state maintains the value of 2.15 mA/cm$^2$, which remains 98% of the initial value before bending (Supplementary Fig. 11b). Also, our ultraflexible OPD shows excellent storage stability up to 100,000 min under ambient air (Supplementary Fig. 12). The excellent air stability proofs a favourable characteristics of our ultraflexible OPD for self-powered devices. The OPD and PLED were used for the PPG sensor in further experiments.

**Self-powered ultraflexible PPG sensors**. A sufficient output power from the OPV is required to drive the PLED and make the PPG sensor self-powered. In order to increase the output power and voltage of OPV single cells, OPV modules containing several OPV single cells were designed and optimised. To optimise the OPV module design, the output current and voltage of the OPV module for varying numbers of series connections were calculated from the single-cell characteristics. The output current and voltage were calculated as $I_{OUT} = 0.8 \times \frac{I_{SC}}{n}$, $V_{OUT} = 0.8 \times V_{OC} \times n$, in which $I_{SC}$ is the short-circuit current of the single cell; $V_{OC}$ is the open-circuit voltage of the single cell; and $n$ is the number of series-connected single cells that constitute the module. $J_{SC}$ and $V_{OC}$ are multiplied by 0.8 to compromise typical value of fill-factor as 0.64 of OPV modules. Making the output current and voltage of the OPV modules equal to that of the PLED is crucial for efficient operation of self-powered PPG sensors. The results of the calculation indicated that OPV modules with 10 series-

connected single cells have the highest overlap with the I–V characteristics of the PLED (Supplementary Fig. 13). Based on this observation, OPV modules with series numbers of 6, 8, and 10 were fabricated (Supplementary Fig. 14a–c). In the OPV modules, $I_{OUT}$ decreased and $V_{OUT}$ increased proportionally as the increase in the number of series connections. As the maximum output powers of all the OPV modules became similar, the voltage required to obtain the maximum power point (MPP) increased proportionally with an increase in the number of series connections (Supplementary Fig. 14d). The constant output power characteristics were examined by keeping the total area of the module, which is the area of individual cells multiplied by series number, constant (1.2 cm$^2$) for every OPV module in this experiment. The operational point of the PLED powered by the OPV module was examined by overlaying the reversed I–V curves of PLED with the I–V curves of the OPV module (Supplementary Fig. 14e). From the results, the number of series connections required for the optimal OPV module design was calculated to be 10. This finding was similar to the result obtained from the previous calculation using single-cell properties; hence, we concluded that having 10 series connections is optimal for PLED operation. Hence, OPV modules with 10 series connections were fabricated on an ultraflexible substrate (Supplementary Fig. 15a, b). These ultraflexible OPV modules exhibited a PCE of 5.8%, which is comparable to the PCE of the module on the glass substrate (6.5%) (Supplementary Table 1). The operational

stability of our OPV modules under air is also important. The ultraflexible OPV cells with same material showed 90% of initial PCE even after 3 h continuous 1-sun illumination with MPP tracking test under ambient air, which indicates our OPV module should also has comparably stable characteristics under MPP tracking[52].

Subsequently, PLED operation with a 10 series-connected OPV module was experimentally examined. Simulated sunlight with a neutral density filter and with various optical densities was used as a variable-intensity input light source. A photograph depicting PLED operation with the OPV, and a schematic of the electrical circuit with the PLED and OPV are shown in Supplementary Fig. 16a, b, respectively. The light-intensity dependences of the OPV modules were measured and overlaid with the PLED $I–V$ curve (Supplementary Fig. 16c). From the results of Supplementary Fig. 16c, circuit current of PLED and OPV were expected to increase with the increase in the input light intensity. Supplementary Fig. 16d shows the experimentally measured dependence of the circuit current and PLED luminance on the light intensity. As expected, circuit current and luminance proportionally increased with the increase in the light intensity. The PLED was observed to emit bright light with an input light intensity of about $5000 \mu W/cm^2$, which is 0.05 times the intensity of sunlight (Supplementary Fig. 16e). The PCE of OPV under indoor light shows 28.1% with 1000 lux of fluorescent lamp illumination[53]. This value is equivalent as power output of $78.2 \mu W/cm^2$ whereas power output under $5000 \mu W/cm^2$ simulated sunlight is $290 \mu W/cm^2$. With these results, our self-powered PPG system should work under 1000 lux of indoor light with OPV modules of $4.5 cm^2$ device area.

Finally, the operation of the self-powered PPG sensor was demonstrated. The operation mechanism of a PPG sensor is shown in Fig. 4a. When light is emitted from the PLED and penetrates the finger, a part of the light is reflected at the blood vessels, while the remaining part is absorbed into the blood. The volume of blood vessels changes with every blood pulse, and the intensity of the reflected light is consistent with the volume pulsation of the blood vessel. The intensity of the reflected light can be detected by an OPD placed on the finger, and the blood pulse rate can be calculated from the OPD signals[5]. First, a self-powered PPG sensor consisting of an OPD, PLED, and OPV module on a glass substrate was fabricated and examined. The PLED was powered by an OPV module with 10 series connections, and the OPV module was illuminated with simulated sunlight of one-sun intensity. The PLED switched on when the OPV module was exposed to the sunlight, and it switched off when the OPV module was stored in the dark. Supplementary Fig. 17 shows the OPD signals when the PLED light was switched on and off by intermittently exposing the OPV module to the simulated sunlight. When the PLED light switched off, the OPD showed an almost constant voltage, as indicated by the black line in Supplementary Fig. 17. However, the OPD showed a clear periodic peak at 1.2 Hz, which is the frequency of the blood pulse, when the PLED was turned on. As shown in Fig. 4b, our PPG sensor performs very stable signal detection, even when the measurement duration is 20 s. The raw data of PPG signal without signal filtering also shows a large artifact in the low frequency (Supplementary Fig. 18). Also, there should be certain artifact from the body and finger movement. Therefore, there will be a certain challenge in the external circuit to process signal during the measurement and to monitor the signal continuously for the long-term measurement.

After the self-powered PPG detection was confirmed to be operational on the glass substrate, self-powered PPG sensors on an ultraflexible substrate were fabricated. For this fabrication, previously reported lamination methods with 6-μm-thick adhesive tape[28] were used to combine the ultraflexible OPV module, OPD, and PLED (Supplementary Fig. 19). A photograph of the ultraflexible, self-powered PPG sensor is shown in Fig. 4c. Finally, the PPG signals from the self-powered ultraflexible PPG sensor were measured (Fig. 4d). Based on the results, blood pulse frequency was calculated to be 77 bpm. Although the periodic signal of the OPD was visible in the result, a large noise in the detected signal was also observed. Some of the reasons for the high noise in the ultraflexible PPG sensor were the large leak current of the ultraflexible OPD and the weak light intensity of the ultraflexible PLED due to the limited power supply from the ultraflexible OPV module.

The reason of poor signal quality with ultraflexible PPG sensors might be the deformation of the ultraflexible sensors. Since our finger was placed on top of the PPG sensors in the measurement, the ultraflexible sensor was deformed by the fingers. The deformation of the sensors will result in both electrical and optical noises, such as the incoming angle of reflected light, the reflected light path inside of the skin, and the electrical noise from the device deformation, which would lead poor signal quality of PPG. The active voltage regulator (AVR) circuit would be critical for stable power output in self-powered devices[54]. Since the AVR has an input voltage range from 7 to 15 V, the AVR have a feasibility to be combined with our 10 series-connected OPV module (8.2 V output) and regulate the voltage to 6.5 V for stable power output. Another critical requirement for self-powered device is the auxiliary battery, especially for their stable operation in dark situation. In Supplementary Fig. 20, the circuit with auxiliary rechargeable battery are explained. While the OPV module differs their current and voltage by the light intensity, the battery has stable current and voltage output and thus, the stable output light of PLED will be achieved. 180-μm-thick battery[55] and 40-μm-thick supercapacitor[56] would be used to achieve self-powered PPG sensors with auxiliary battery. Connecting solar cells to PLEDs without AVR or auxiliary battery will create a variable drive current in the PLEDs, that might result variable output of OPDs, and the unreliable bio-signal detection. Lastly, signal sampling is a critical problem for state-of-the-art flexible, self-powered electronics. In both an ultraflexible self-powered organic electrochemical transistor[57] and a flexible self-powered PPG sensor[56], the sampling systems were rigid ICs and separately connected as external circuits. Though our OPV module is capable to operate signal monitoring ICs because their power consumption is ~0.2 μW (TWILITE, Mono Wireless), certain effort is necessary to make these ICs ultraflexible and integrate to the self-powered systems.

## Discussion

In this study, a self-powered, ultraflexible PPG sensor employing air-operation-stable, ultraflexible PLEDs was fabricated. To the best of our knowledge, this is the first study that demonstrates the use of self-powered optoelectrical sensors in ultraflexible organic devices. This self-power technology used in this study will pave the wave of ultraflexible wearable optoelectronic devices which take an important role for future ubiquitous healthcare society.

## Methods

**Materials**. The PTzNTz-BOBO samples were synthesised from microwave reaction with thiazolothiazole monomer and dibrominated naphthobisthiadiazole monomer[58]. $PC_{71}BM$ was purchased from Solenne BV Corporation. PEIE, Liq, Zinc acetate dihydrate, 2-methoxyethanol, and 2-aminoethanol were purchased from Wako Chemical. The yellow light-emitting polymer, SY, was purchased from Sigma-Aldrich. The glass substrate with a patterned ITO electrode was purchased from EHC Ltd. The sheet resistance of ITO was <10 Ω/sq.

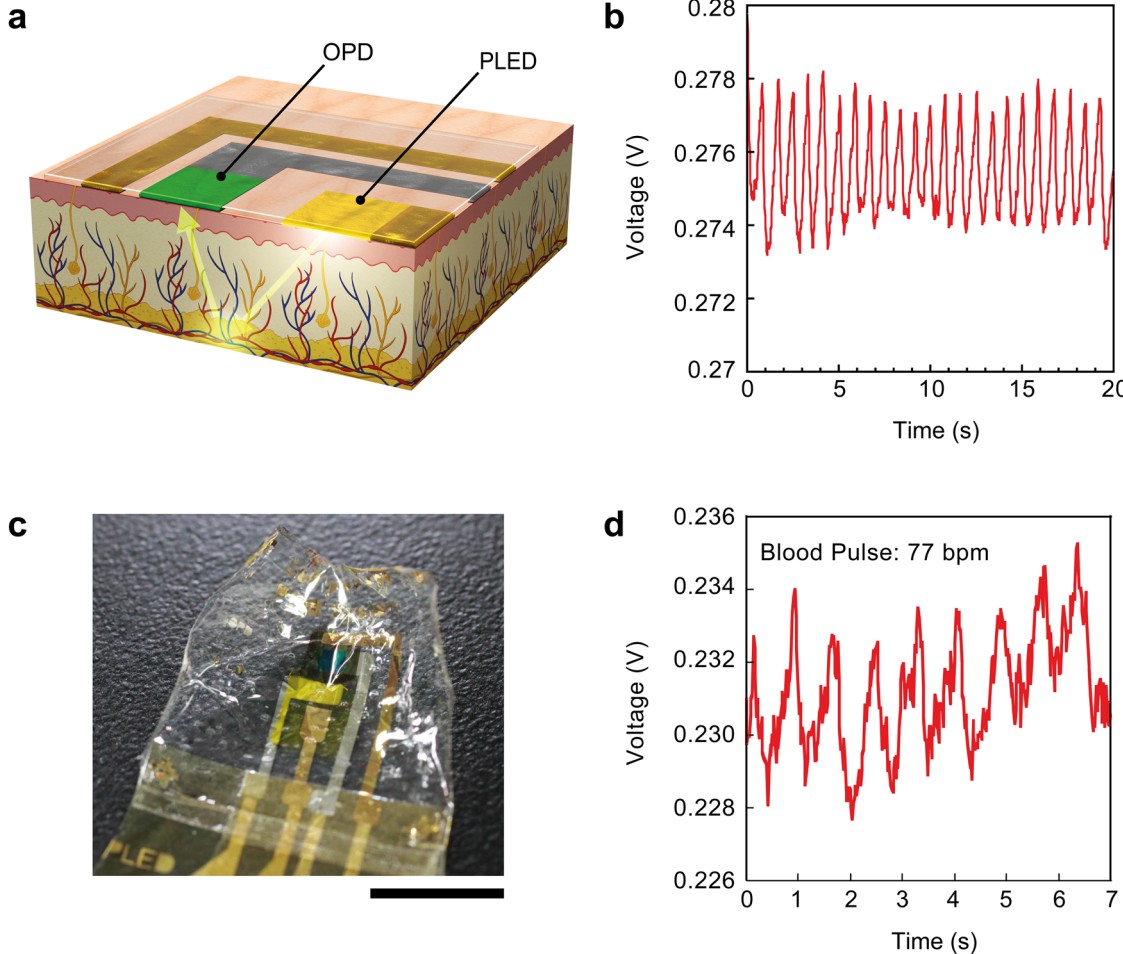

**Fig. 4 Blood pulse detection of self-powered photoplethysmogram sensors. a** Schematic diagram describing the operational mechanism of photoplethysmogram (PPG) by polymer light-emitting diode (PLED) and organic photodiode (OPD). **b** Output voltage characteristics of OPD with PPG measurement. The PLED was powered by 10 series-connected organic photovoltaic (OPV) modules with one-sun illumination from a solar simulator. All devices are on glass substrate. **c** A photograph of the ultraflexible, self-powered PPG sensor. Scale bar indicates 1 cm. **d** Output voltage characteristics of OPD with PPG measurement. The PLED was powered by 10 series-connected OPV modules with one-sun illumination from a solar simulator. All devices are on ultraflexible substrate. Blood pulse frequency was calculated to be 77 beats per minute (bpm) from the measurement results.

**Ultraflexible film substrate fabrication.** The ultrathin film substrates were composed of double-layered parylene (diX-SR, Daisan Kasei Co., Ltd.) and epoxy (SU-8 3005, MicroChem). First, a 1-μm-thick parylene layer was deposited via chemical vapour deposition (CVD) on a glass substrate whose surface was coated with a fluorinated polymer (Novec 1700, 3M). A 500-nm-thick epoxy layer was then spin-coated (5000 rpm for 60 s) on a 1-μm-thick parylene layer as the planarisation layer. The film was then annealed at 95 °C for 3 min after ultraviolet exposure. Subsequently, the substrates were annealed under a nitrogen atmosphere at 180 °C for 30 min. All ultrathin devices were electrically connected with thin Au wiring on a 12.5-μm-thick polyimide substrate using anisotropic conductive film (3M, ECATT 9703) tapes as contact parts.

**Inverted PLED fabrication.** A 100-nm-thick ITO layer was subsequently formed as the transparent electrode by DC sputtering. The ITO electrodes were patterned via photolithography using a ZPN (Nihon Zeon) negative resist and an ITO-07N (Kanto Chemical) ITO etcher. A 20-nm-thick ZnO layer was used as the ETL. The substrates were spin-coated with a ZnO precursor (5000 rpm for 30 s) solution prepared by dissolving zinc acetate dehydrate (0.5 g) and ethanolamine (0.16 mL) in 5 mL 2-methoxyethanol[59]. The substrates were then baked in air at 180 °C for 30 min. To form PEIE:Liq layer, the mixture of 1.5 wt% solution of PEIE and Liq with 2-methoxyethanol was deposited on a substrate via spin-coating (4000 rpm 60 s). After annealing the substrate at 100 °C for 1 min, the device was rinsed with ethanol and annealed at 100 °C for 1 min again. The SY layer was deposited via spin-coating using a toluene solution of 6 mg/ml. Subsequently, the MoO$_X$ (10 nm) and Al (80 nm) layers were deposited via vacuum evaporation. Finally, a 1-μm-thick parylene layer was deposited via CVD to form a passivation layer.

**OPV and OPD fabrication.** A ZnO layer was deposited using the same methods as that used for the PLED. PTzNTZ-BOBO and PC$_{71}$BM were mixed as a 1:1.5 of weight ratio. Then, chlorobenzene was added to a mixture with 5 g L$^{-1}$ PTzNTZ-BOBO concentration. The solution was stirred at 100 °C for 1 h, and 1 vol% of 1,8-Diiodooctane was added based on the amount of chlorobenzene after the temperature of the solution decreased back to 20–30 °C in under N$_2$ atmosphere. The solution was spin-coated on the device in a glove box at 600 rpm for 20 s. The active layer thickness is 200 nm. MoO$_X$ (7.5 nm) and Ag (100 nm) were sequentially deposited via thermal evaporation, under a pressure of less than 1 × 10$^{-3}$ Pa, as a top interlayer and an anode, respectively. Finally, a 1-μm-thick parylene layer was deposited via CVD to form a passivation layer.

**Device characterisation.** The OPV modules with a total cell area of 1.2 cm$^2$ and OPD with an active area of 0.04 cm$^2$ were characterised under one-sun illumination using a solar simulator (AM 1.5 global spectrum with 1000 W m$^{-2}$ intensity calibrated using a silicon reference diode, Bunkokeiki). The J–V characteristics were recorded using a Keithley 2400 source metre for every 10 mV in an ambient laboratory atmosphere, without humidity or temperature control (with an approximate temperature of 20 °C and humidity of 30% relative humidity (RH)). Accordingly, external gold wirings were used for the electrical contacts of the OPVs on the ultrathin substrates. The 100-nm-thick gold wirings were deposited under a vacuum through a shadow mask onto 12.5-μm-thick polyimide films. One side of the wirings were connected to the electrodes on the freestanding foils using an electrically conductive adhesive transfer tape (3 M™, ECATT 9703). The other sides were connected to the source metre using alligator clips. Light power of the PLED was calibrated with optical power metre (1936-R, Newport).

To evaluate the PLED characteristics, a light distribution measurement system (C9920-11, Hamamatsu Photonics) and an EQE measurement system (C9920-12, Hamamatsu Photonics) were used. For the air-stability tests, the devices were operated under ambient air (~20 °C, 30% RH). The measurements were also performed in an ambient atmosphere.

To perform the PLED operation experiment with the OPV as a power source, a solar simulator and an optical filter were used to modulate the light intensity for simulated sunlight input. A schematic of the experimental setup is shown in Supplementary Fig. 21. Anode of PLED and OPV module are electrically connected. Cathode of the PLED and the OPV module are connected each other through digital multi metre.

The study protocol was thoroughly reviewed and approved by the ethical committee of the University of Tokyo (approval no. KE18-41).

**Statistics and reproducibility**. All devices present this paper were fabricated in several times and show reproduced results by several persons. Note that connecting solar cells to PLEDs without voltage regulator or auxiliary battery will create a variable drive current in the PLEDs, that might result variable output of OPDs, and the unreliable PPG measurement in Fig. 4d.

## Data availability
All relevant data present in this paper are available from the corresponding author upon reasonable request.

## Code availability
All relevant codes present in this paper are available from the corresponding author upon reasonable request.

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

## Acknowledgements

This work was financially supported by Japan Science and Technology Agency ACCEL grant number JPMJMI17F1. Authors thank TECOLAB INC. to support creating images in Figs. 1 a, b and 4 a.

## Author contributions

H.J., T.Y., K.F., and T.S. conceived and designed the research. M.S. and I.O. synthesised and provided photovoltaic materials. H.J., M.K., and W.Y. fabricated ultraflexible devices and performed electrical and optical measurements. K.F. and T.Y. analyzed and interpreted the data. The manuscript was prepared with comments from all of the co-authors. T.S. supervised the project.

## Competing interests

The authors declare no competing interests.
