## [Peer Review File · Nature Communications]

Reviewer #1 (Remarks to the Author):

The authors have satisfactorily addressed most of my comments in the previous review report (for their original submission to Nature Electronics), except the following two points. I would recommend the publication of this work on Nature Communications if the authors could properly address these two issues.

1. Regarding the poor quality of the PPG signal, the authors attributed it to the dark current increase of the OPD due to the rigidity and roughness of the ZnO layer used in the inverted structure. However, the OPD they fabricated is already ultraflexible, and the rigidity of a single layer shouldn't be a concern. The authors should further clarify the actual reason for the poor signal quality.
2. The author should specify the light intensity used to measure the rise and fall time of the OPD. The response speed of the organic photodetector may drop significantly at low light intensity.

Reviewer #2 (Remarks to the Author):

I thank the authors for the detailed response. While the rebuttal addressed most of the formatting concerns and missing information, the biggest two still remain.

1. The major problem is the non-diode behavior of the device in Fig. 2c, it doesn't show a turn-on, hence, it's not a light-emitting "diode". While the authors explained the ZnO layer surface roughness is the culprit. It will be good to provide the same characterization for the glass device to convince readers that ZnO is problematic in the ultra-flexible device, while it is not for the glass device.
2. Connecting solar cells to LEDs without power regulation and storage will create a variable drive current in the LEDs, that will change your current output from the OPD. Therefore, the bio-signal will be unreliable. Please ensure this is stated in the caption of the figure as well as the manuscript.

Reviewer #3 (Remarks to the Author):

The new version of the manuscript has addressed all my concerns in my original report, and the additional results and analyses further improve the quality of this paper. I fully support publication of this paper as is.

Point-to-point Reply to Reviewers' Comments

=====

Author's reply to Reviewer #1

=====

Reviewer #1 (Remarks to the Author):

Comment #1-1

The authors have satisfactorily addressed most of my comments in the previous review report (for their original submission to *Nature Electronics*), except the following two points. I would recommend the publication of this work on *Nature Communications* if the authors could properly address these two issues.

Reply #1-1

We are grateful for Reviewer #1's thoughtful comments. Based on the upcoming comments, our manuscript is revised to answer the two issues mentioned below.

Comment #1-2

Regarding the poor quality of the PPG signal, the authors attributed it to the dark current increase of the OPD due to the rigidity and roughness of the ZnO layer used in the inverted structure. However, the OPD they fabricated is already ultraflexible, and the rigidity of a single layer shouldn't be a concern. The authors should further clarify the actual reason for the poor signal quality.

Reply #1-2

We agree with this comment. The reason of poor signal quality with ultraflexible PPG sensors might be the deformation of the ultraflexible sensors. Since our finger was placed on top of the PPG sensors in the measurement, the ultraflexible sensor was deformed by the fingers. The deformation of the sensors will result in both electrical and optical noises, such as the incoming angle of reflected light, the reflected light path inside of the skin, and the electrical noise from the device deformation, which would lead poor signal quality of PPG.

Corresponding sentences are added to the manuscript in Line 329-334, Page 19.

Comment #1-3

The author should specify the light intensity used to measure the rise and fall time of the OPD. The response speed of the organic photodetector may drop significantly at a low light intensity.

Reply #1-3

We agree with this comment. The light intensity used to measure OPD rise and fall time (8

mW/cm² at a wavelength of 550 nm) is added to **the figure caption of Supplementary Fig. 10.**

=====

Author's reply to Reviewer #2

=====

Reviewer #2 (Remarks to the Author):

Comment #2-1

I thank the authors for the detailed response. While the rebuttal addressed most of the formatting concerns and missing information, the biggest two still remain.

Reply #2-1

We appreciate all of the comments from Reviewer 2. The manuscript has been revised with reflecting all of the comments and we try to convey every answer point by point.

Comment #2-2

The major problem is the non-diode behavior of the device in Fig. 2c, it doesn't show a turn-on, hence, it's not a light-emitting "diode". While the authors explained the ZnO layer surface roughness is the culprit. It will be good to provide the same characterization for the glass device to convince readers that ZnO is problematic in the ultra-flexible device, while it is not for the glass device.

Reply #2-2

We agree with this comment. Based on this comment, we conducted the AFM experiment for the ZnO surface on the glass substrate (Supplementary Fig. 9). As shown in Supplementary Fig. 9 b and c, the ZnO layer on top of the ultraflexible substrate has rough surface while that on the rigid glass substrate shows a smoother surface. The difference is indicated as Rms, which on ultraflexible substrate has 12.5 nm and on glass substrate has 7.79 nm, respectively.

Supplementary Figure 9 | Surface Images of PLED interface layers with Atomic Force Microscope (AFM). Surface images of **a**, ITO electrode on top of 1.5 μm -thick ultraflexible substrate, **b**, ZnO interface layer on top of ITO and ultraflexible substrate, **c**, ZnO interface layer on top of ITO and glass substrate, and **d**, ZnO:Liq interface on top of ITO and ultraflexible substrate. Rms of ZnO layer on ultraflexible substrate is 12.5 nm and that of ZnO layer on glass substrate is 7.79 nm, respectively.

Corresponding figures and sentences are added to the manuscript in Supplementary Fig. 9 b, and Line 188-191, Page 11.

Comment #2-3

Connecting solar cells to LEDs without power regulation and storage will create a variable drive current in the LEDs, that will change your current output from the OPD. Therefore, the bio-signal will be unreliable. Please ensure this is stated in the caption of the figure as well as the manuscript.

Reply #2-3

We agree with this comment. Reflecting this comment, the sentence about the unreliability of obtained bio-signal is added to **the figure caption of Figure 4 and the manuscript in Line 351-353, page 20.**

=====

Author's reply to Reviewer #3

=====

Reviewer #3 (Remarks to the Author):

Comment #3-1

The new version of the manuscript has addressed all my concerns in my original report, and the additional results and analyses further improve the quality of this paper. I fully support publication of this paper as is.

Reply #3-1

We thank Reviewer 3's positive comment for publishing our manuscript to *Nature Communications*. The manuscript is revised again based on the comments received from other reviewers. It is grateful if you could check about the revised point from your side.

Reviewer #1 (Remarks to the Author):

The authors have satisfactorily addressed my comments. I recommend the acceptance of this manuscript.

Reviewer #2 (Remarks to the Author):

While I am not still convinced regarding the source of the non-diode behavior of the ultraflexible PLED, I am recommending the publication of this article, which will allow the authors and others in the research community to work on this problem.